# Identification of Candidate Genes Related to the Husk Papillae in Foxtail Millet (*Setaria italica* (L.) P. Beauv)

**DOI:** 10.3390/plants14162535

**Published:** 2025-08-14

**Authors:** Meixia Tan, Yang Yang, Zhe Chen, Xiangyuan Gong, Fangfang Ma, Ming Duan, Lidong Wang, Yuanhuai Han

**Affiliations:** 1College of Agriculture, Shanxi Agricultural University, Taiyuan 030031, China; tanmeixia186@163.com (M.T.); yang116@sxau.edu.cn (Y.Y.); cz18406462801@163.com (Z.C.); 13994174295@163.com (X.G.); mff1984abc@163.com (F.M.); 2Experimental Teaching Center, Shanxi Agricultural University, Taiyuan 030031, China; duanming840305@163.com; 3Department of Basic Sciences, Shanxi Agricultural University, Taiyuan 030031, China; wanglidong@sxau.edu.cn

**Keywords:** foxtail millet, husk, papillae, BSA-Seq, SNPs, InDels

## Abstract

Efficient and fast water uptake by seeds, facilitated by optimal soil moisture, plays a critical role in timely germination and early seedling vigor for foxtail millet production in arid and semi-arid regions. The husk, as a unique structure through which the seed contacts the soil, plays an important role in water uptake and germination. Many foxtail millet germplasm accessions have papillae on the epidermis of their husks, yet the role of this trait in water uptake and germination, as well as the genetic basis and regulatory mechanism related to this trait, remain unknown. In this study, we demonstrated that the water uptake by the seeds from accessions with papillae was significantly higher than that of accessions without papillae two hours and four hours after sowing during a 10 h experiment, resulting in faster germination. Analysis of segregating ratios from two F2 populations derived from crossing between accessions with and without papillae indicated that husk papilla density was of monogenic dominance. Bulked Segregant Analysis Sequencing (BSA-Seq) showed that candidate regions on chromosome 5 were significantly associated with husk papilla density. The mapped region overlapped by the two BSA populations for papilla density included 72 genes. In combination with the expression profiles of these genes, five candidate genes were identified, encoding aquaporins, fructose transporter, and glycoside hydrolase. This study elucidated the role of husk papillae in enhancing water uptake and germination in foxtail millet, provided genetic insights into the trait, and laid the foundation for further study on the mechanism of husk papilla differentiation.

## 1. Introduction

Foxtail millet [*Setaria italica* (L.) P. Beauv.] was domesticated from wild green foxtail (*Setaria viridis*) originating from the Yellow River Basin of China approximately 10,000 years ago [1]. As a principal Neolithic crop in North China, it served as the nutritional foundation for the development of ancient Chinese agricultural civilization. Prior to the widespread adoption of rice and wheat during the Han Dynasty (206 BC–220 AD), foxtail millet constituted the staple food crop across East Asia (encompassing China, Japan, and the Korean Peninsula) and significant portions of Eurasia [2]. Notably, this C4 cereal exhibits superior drought tolerance compared to other major grain crops, making it particularly suitable for cultivation in the arid and semi-arid regions of northern China [3,4]. Foxtail millet exhibits exceptional water economy during germination, requiring only 26% of seed weight in water uptake for successful germination, compared to the minimum 45% requirement observed in other cereal crops [5,6]. Water uptake during seed germination is a critical factor for successful seedling establishment, and seed coat structural features (such as surface papillae) may play an important role in this process [7,8,9]. In recent years, multiple studies have demonstrated that the morphological characteristics of seed or fruit coats can significantly influence water uptake capacity. In rapeseed (*Brassica napus*), the number of seed coat papillae shows a significant correlation with water uptake after 5 h of imbibition [10]. Integrating microscopy analyses employing phase-contrast optics and laser micro-interferometry identifies distinct husk epidermal architectures among foxtail millet varieties, with certain accessions exhibiting regularly patterned papilla cells and others displaying smooth epidermal surfaces [11,12]. During the germination stage, the husk forms the primary boundary between the foxtail millet kernel and the soil. However, the roles of papilla cells in the water absorption and germination of foxtail millet seeds, as well as the genetic basis and regulatory sites of this trait, remain unclear at present.

The epidermal specialized cells that have been mostly studied in plants are trichomes. Plant trichomes are widely distributed on the surfaces of flowers, stems, leaves, seed coats, and other tissues [13,14]. Trichomes can not only improve plant tolerance to biotic and abiotic stresses but also secrete secondary metabolites, which have significant medicinal and economic value [15,16,17,18]. The development of plant trichomes is regulated by both genetic and environmental factors, including transcription factors, functional genes, hormones, and epigenetic modifications [19,20,21,22,23,24]. Petal conical cells are another common type of specialized epidermal cells in plants [25], which can focus light more effectively and accelerate the development of flower organs and seeds [26,27]. It also scatters the reflected light from mesophyll cells, which is beneficial for attracting pollinators [28,29,30]. The *MIXTA* gene of *Antirrhinum majus*, the *PhMYB1* gene of *Petunia hybrid*, and the *AtMYB16* gene of *Arabidopsis thaliana* play a role in conical cell development [31,32,33,34]. A new type of epidermal specialized cell, the papilla cell, has been discovered on the leaves or seed coats of some plants [35,36,37,38]. The papillae not only reflect solar radiation but also facilitate water conduction, enabling water to be absorbed by plant cells quickly [10,38]. The *OsRopGEF10* gene plays a significant role in the formation of papillae in rice leaves [36].

There have been differences in the size, density, and arrangement of husk papillae of different foxtail millet varieties [11,12]. To date, there have been no reports on the role of husk papilla cells in the water uptake and germination of seeds and the molecular genetics underlying the formation of papillae in foxtail millet. In this study, we analyzed the differences in water uptake and germination between the varieties with and without papillae. Two F2 populations were constructed between varieties with and without papillae. Using BSA-Seq, one genomic region on chromosome 5 significantly associated with the papilla density was identified. This study may lay the foundation for further research on the molecular mechanism of papilla development and drought tolerance in foxtail millet.

## 2. Results

### 2.1. Analysis of Water Uptake and Germination of Seeds with and Without Papillae on the Husks

Investigation of the papilla density among germplasms revealed that the papilla density of JinGu36 (JG36) and JinGu28 (JG28) was high, whereas HuaJin3 (HJ3) and HuaJinkang2 (HJK2) had no papillae on the husk. To study the relationship between husk papillae and water uptake and germination, we analyzed the differences in water uptake and germination between the variety JG36 with papillae and the variety HJ3 without papillae, as well as between the variety JG28 with papillae and the variety HJK2 without papillae at different time intervals. It was calculated that the water uptake of JG36, HJ3, JG28, and HJK2 reached 17%, 16%, 17%, and 14% two hours after sowing (Figure 1, Appendix A). The water uptake of JG36 and JG28 was significantly higher than that of HJ3 and HJK2 two hours and four hours after sowing. The germination of JG36, HJ3, JG28, and HJK2 reached 7%, 2%, 25%, and 0%, respectively, twenty hours after sowing (Figure 1, Appendix A). The germination of JG36 and JG28 was significantly higher than that of HJ3 and HJK2, respectively, twenty to twenty-nine hours after sowing.

### 2.2. Genetic Analysis of Papillae on the Husk in Foxtail Millet

To investigate the inheritance of papillae on the husk, two hybrid populations were constructed in this experiment. The husks of JG36 exhibited numerous small papillae, whereas those of HJ3 were smooth and devoid of papillae (Figure 2). The F1 seeds from the cross between JG36 and HJ3 displayed papillae on the husks. Phenotypic data were collected from the two parental lines, the F1 hybrids, and the two F2 populations (Table 1). All 15 F1 plants exhibited papillae on the husks, consistent with the phenotypes of the parental lines JG36. In the JG36 × HJ3 F2 population (164 plants), 128 individuals exhibited papillae on the husks, while 36 lacked papillae. The observed segregation ratios were tested against the expected 3:1 ratio using a χ2 goodness-of-fit test for F2 populations. The ratio of papilla to non-papilla plants closely matched the expected 3:1 segregation ratio. It indicated that the papilla density was regulated by a single dominant gene.

Similarly, the husks of JG28 exhibited numerous small papillae, whereas those of HJK2 were smooth and devoid of papillae (Figure 3). All F1 seeds from the cross between JG28 and HJK2 displayed papillae on the husk. Phenotypic data were collected from the two parental lines, the F1 hybrids, and the two F2 populations (Table 2). All 15 F1 plants exhibited papillae on the husks, consistent with the phenotypes of the parental line JG28. In the JG28 × HJK2 F2 population (182 plants), 139 individuals exhibited papillae on the husks, while 43 lacked papillae. The ratio of papilla to non-papilla plants closely matched the expected 3:1 segregation ratio. It also indicated that the papilla density was regulated by a single dominant gene.

### 2.3. Loci of Papillae on the Husk Identified Based on BSA-Seq

Bulked Segregant Analysis Sequencing (BSA-Seq) of two F2 populations was conducted to identify and map the genetic loci associated with the papilla density. Genomic DNA from four parental lines (JG36, JG28, HJ3, and HJK2) and four bulks (two papilla bulks and two non-papilla bulks) was sequenced using an Illumina HiSeq 2500 platform. The clean data of JG36, JG28, HJ3, and HJK2 obtained by high-throughput sequencing generated approximately 30.7 Gb, 52.9 Gb, 29.7 Gb, and 48.7 Gb, respectively (Table 3). The Q30 values for all eight samples exceeded 92.00%, indicating high-quality sequencing data. All sequencing reads were aligned to the Yugu1 (T2T) reference genome, with over 95.00% of clean reads successfully mapped for each sample. For the JG36 × HJ3 F2 population, a total of 2,002,017; 719,803; 2,113,196; and 2,092,266 polymorphic SNPs were identified in JG36, HJ3, the papilla bulk, and the non-papilla bulk, respectively. Similarly, 255,095; 98,418; 283,084; and 281,108 small InDels were identified in JG36, HJ3, the papilla bulk, and the non-papilla bulk, respectively. For the JG28 × HJK2 F2 population, a total of 2,339,553; 1,297,415; 2,567,165; and 2,537,469 polymorphic SNPs were identified in JG28, HJK2, the papilla bulk, and the non-papilla bulk, respectively. Additionally, 294,988; 172,874; 340,174; and 338,819 small InDels were identified in JG28, HJK2, the papilla bulk, and the non-papilla bulk, respectively.

Based on the genotyping results, 1,095,293 homozygous differences in SNP sites between the two parents, JG36 and HJ3, were screened. With JG36 as the reference parent, we calculated the SNP-index values of 1,095,293 marker sites for the two bulks. When all individuals in the bulk were matched to the reference genotype, they were assigned a value of 0; when they were completely mismatched, they were assigned a value of 1. The ΔSNP-index of each site was obtained by subtracting the SNP-index values of the papilla bulk from the SNP-index values of the non-papilla bulk. The number of ΔSNP-indexes on each chromosome was calculated, and the distribution of the ΔSNP-indexes was mapped on the chromosomes. The absolute value of the ΔSNP-index was equal to 0.5 as the threshold line for screening, and windows larger than the threshold were selected as the candidate interval. The SNPs within the candidate interval were annotated using the reference genome of the foxtail millet Yugu1 (T2T) annotation file. The number of ΔSNP-indexes on chromosome 5 was the largest (Figure 4a). The locus with the largest absolute value of the ΔSNP-index was on chromosome 5. There were 127 genes within the candidate interval. Meanwhile 175,563 homozygous differences in InDel sites between the two parents JG36 and HJ3 were screened. Using the same method, we calculated the ΔInDel-index of each site between the two bulks. The number of ΔInDel-indexes on each chromosome was calculated, and the distribution of the ΔInDel-indexes was mapped on the chromosomes. The absolute value of the ΔInDel-index was equal to 0.5 as the threshold line for screening. The number of ΔInDel-indexes on chromosome 5 was the largest (Figure 4b). The locus with the largest absolute value of the ΔInDel-index was on chromosome 5. There were 96 genes within the candidate interval.

Similarly, based on the genotyping results, 1,147,050 homozygous differences in SNP sites between the two parents, JG28 and HJK2, were screened. With JG28 as the reference parent, we calculated the SNP-index values of 1,147,050 marker sites for the two bulks. The number of ΔSNP-indexes on chromosome 5 was the largest (Figure 5a). The locus with the largest absolute value of the ΔSNP-index was on chromosome 5. There were 109 genes within the candidate interval. Meanwhile, 190,207 homozygous differences in InDel sites between the two parents, JG28 and HJK2, were screened. The number of ΔInDel-indexes on chromosome 5 was the largest (Figure 5b). The locus with the largest absolute value of the ΔInDel-index was on chromosome 5. There were 74 genes within the candidate interval.

### 2.4. BSA Co-Localization Analysis Based on Two F2 Populations

There were 63 common genes found between these two BSA populations through ΔSNP-index association analysis (Figure 6). Through ΔInDel-index association analysis, 37 common genes were found between these two BSA populations. As some genes appeared in both groups, a total of 72 genes were obtained (Appendix A).

### 2.5. Expression Analysis of Candidate Genes for Papilla Density at Different Developmental Stages

Because papilla cells belong to the specialized cells of the epidermis of the husk, we conducted transcriptome sequencing on the husk of JG36, a material with extremely high papilla density, at different developmental stages. The expression patterns of candidate genes for papilla density were analyzed (Appendix A). Among the 72 genes, 33 genes were not expressed in the husks at all four stages (Figure 7). There were 26 genes that were highly expressed in the husks at all four stages. Among the remaining 13 genes, *Seita.5G009400*, *Seita.5G010100*, *Seita.5G015400*, *Seita.5G019600*, *Seita.5G021300*, and *Seita.5G023200* were highly expressed in the husks of the last two stages. *Seita.5G020100* and *Seita.5G025400* were lowly expressed in the husks of the first three stages. *Seita.5G007400*, *Seita.5G007500*, *Seita.5G021600,* and *Seita.5G022400* were highly expressed in the husks of the first two stages. *Seita.5G007300* was highly expressed in the husks of the second stage (Appendix A).

### 2.6. Functional Prediction of Candidate Genes for Papilla Density

During the third stage of husk development, that is, the pollination stage, the papilla cells on the husks have been fully differentiated and formed; the candidate genes regulating papilla formation should be highly expressed during the first two stages. A total of five candidate genes were found to be highly expressed in the husks of the first two stages (Appendix A). Their homologous genes in rice were found in the Phytozome 13 database, and the functional annotation information of homologous genes was searched in the China Rice Data Center to preliminarily predict the functions of the candidate genes. According to the annotation of homologous genes, the five candidate genes were named *SiTIP4;2.1*, *SiTIP4;2.2*, *SiTIP4;2.3*, *SiSWEET16*, and *SiGH9B16.* (See Table 4).

## 3. Discussion

Seed germination is a crucial stage for plant growth and development, and it is also one of the stages most sensitive to drought stress [43,44]. Foxtail millet is generally grown in rain-fed arid regions, where fast germination is critical for plant population establishment. In this study, we demonstrated that the water uptake by foxtail millet seeds was extremely rapid, with water content reaching approximately 30% (Appendix A and Table 3) within just 2 h after sowing. This exceeds the threshold (26%) required for germination [5,6]. What’s more, accessions with papillae showed significantly higher water uptake than those without two hours and four hours after sowing. In oil rape, 35 accessions were categorized into different groups based on the density and size of the papilla on the seed coat; accessions with more and bigger papillae or smaller papillae showed more rapid imbibition than those with fewer and smaller papillae [10]. The high density of specialized structures, such as husk or seed coat papillae, may have been selected during domestication to enable rapid germination, thus ensuring robust seedling establishment in rain-fed arid regions. The presence of papillae would increase the surface area and may also create microstructures on the husk surface that enhance hydrophilicity or capillary action, thereby promoting initial water uptake. Therefore, increasing the papilla density on the husks of foxtail millet could be an important way to improve germination and guarantee the required population density for a stable and high yield, especially in rain-fed regions.

In addition to the fast water uptake promoted by papillae on the husk for timely germination, these microstructures may also play roles in avoiding feeding by birds and seed dispersal. Similar to the adaptive divergence in Acmispon wrangelianus seed color camouflage driven by selection from seed predators on contrasting soils [45], the papillae-induced differences in light reflection on foxtail millet husks may represent an alternative mechanism for reducing detection by avian or insect seed predators through visual deception. Secondary dispersal after seeds fallen from the plants by overland flow is critical in drylands as emphasized by Thompson et al. and their BOB-CELC model [46]. Our findings suggest that papillae on foxtail millet husks create surface roughness, probably significantly enhancing seed adhesion to sediment in runoff and promoting seed dispersal. In addition, papillae on the husk may make the seeds more likely to be adhered to animal feet for dispersal.

Through the combined analysis of BSA and transcriptome, we identified five candidate genes related to papilla density. Among the candidate genes, the three genes *SiTIP4;2.1*, *SiTIP4;2.2*, and *SiTIP4;2.3* showed homology to the aquaporin gene *OsTIP4;2* in rice. The aquaporins are a type of channel protein widely present on biological membranes, which can efficiently and selectively transport water molecules and small molecule solutes (such as glycerol, urea, etc.) and play a key role in plant water balance, cell elongation, and environmental response [47]. Aquaporin proteins control the formation of multicellular hairy bodies in tomatoes by regulating the water absorption and turgor pressure changes in the hairy body precursor cells [39]. During the elongation process of cotton fibers, aquaporin promotes the rapid elongation of cotton fiber cells by regulating the cellular water potential and osmotic balance [40]. Therefore, these aquaporin genes could be involved in regulating the husk papilla density in foxtail millet.

*SiSWEET16* encodes the fructose transporter homologous to *OsSWEET16* in rice. Hexoses are known to play signaling roles in development [48]. In cotton, the expression of *vacuole invertase* (*VIN*) is an important early event in the formation of cotton fibers [41]. *VIN* may regulate cytosolic hexose levels and sugar homeostasis by coupling with the activity of tonoplast sugar transporters [48], such as the tonoplast fructose exporter *AtSWEET17* [49] and the glucose exporter *AtERDL6* [50], thereby regulating cotton fiber initiation. It is then possible that *SiSWEET16* may play a role in the differentiation of papillae on the husks in foxtail millet.

Another candidate gene, *SiGH9B16,* encodes the glycoside hydrolase family protein homologous to *OsGH9B16* in rice. Glycoside hydrolases are a type of enzyme capable of catalyzing the hydrolysis of glycosidic bonds and are widely involved in processes such as plant cell wall metabolism, sugar signal regulation, and defense responses [51]. In *Arabidopsis*, endo-1,4-beta-D-glucanases, *KOR2*, and *KOR3*, are related to root hair initiation and trichome formation [42].

Based on the functional annotation of the candidate genes, either related to water transportation, sugar production, or transportation, and the roles of their homologous genes in other plants, it is inferred that they may be involved individually or in combination in the regulation of husk papillae in foxtail millet. Further experiments are needed to clarify how these candidate genes might regulate the differentiation and formation of husk papillae.

Further investigation of the occurrence/frequency of the husk papillae among the germplasms of *S. italica* and its wild relative *S. viridis* could reveal if the trait is selected during domestication or if it is selected by breeders for different ecological regions, especially those with contrasting rainfall during the sowing season.

## 4. Materials and Methods

### 4.1. Plant Materials

Four foxtail millet varieties, JG36, JG28, HJ3, and HJK2, were used as materials. JG36 and JG28 exhibit small and numerous papillae on their husks (paternal parents), while HJ3 and HJK2 have smooth husks (maternal parents). All four varieties were obtained from and preserved by Shanxi Agricultural University. Two crosses were performed: JG36 was crossed with HJ3, and JG28 was crossed with HJK2. Two F2 populations were generated using the single-seed descent method. The four parental lines and the F2 populations were cultivated in the experimental field of Shanxi Agricultural University (37°25′14″ N, 112°35′25″ E, Taigu, China) during the 2024 growing season (May–October).

### 4.2. Determination of Water Uptake and Germination of Seeds

Equal amounts (5.000 g) of dry seeds (the water content of the dry seeds was about 14% as determined by the FOSS NIRS DS2500, Appendix A) of JG36 and HJ3 were separately weighed. For each variety, 42 samples of equal weight were taken. All 84 samples were simultaneously mixed with 15% wet sand and placed in soil with the same humidity. After 2 h, 4 h, 6 h, 8 h, and 10 h, three samples were taken from each variety, blotted dry, and weighed. Water uptake was then calculated as mean percent weight increase (*n* = 3). Wr/% = (Wa − Wb)/Wb × 100, where Wr is the percent weight increase, Wa is the weight of seeds after contact with water, and Wb is the weight of the dry seed. After 11 h, 14 h, 17 h, 20 h, 23 h, 26 h, 29 h, 32 h, and 35 h, three samples were taken from each variety, blotted dry, and the germination was calculated. The germination was calculated by dividing the number of germinated seeds by the total number of seeds. An analysis of variance (ANOVA) was conducted to determine the significant differences in water uptake and germination among different varieties at different time points. When significant differences were found (*p* < 0.05), Sidak’s multiple comparisons test was used to compare the average values at different time points, maintaining a significance level of 5% (95% confidence). All statistical analyses were performed using GraphPad Prism 6.0 software with a confidence level of 95% (α = 0.05). The method for determination of the water uptake and germination of JG28 and HJK2 seeds were the same.

### 4.3. Phenotypic Data Collection and Analysis

During the late seed-filling stage, three uniformly developed seeds were selected from the middle section of each panicle. The seeds were freeze-dried for 48 h and then mounted on the electron microscope sample stage using conductive tape. Following uniform gold coating of the sample surface, husk surface morphology (including papilla presence/absence) was examined and recorded using scanning electron microscopy. The segregation ratios of papilla and non-papilla phenotypes in the F2 populations were analyzed using the chi-square (χ^2^) tests to compare the observed ratios with the expected Mendelian ratios. A significance threshold of *p* = 0.05 (χ^2^ = 3.84) was applied.

### 4.4. DNA Extraction and Quality Detection

DNA extraction was performed using young leaf tissues from the four parental lines and F2 individuals. The leaf sample of each plant was collected in a 2.0 mL centrifuge tube, labeled with the corresponding plant number, and stored in a refrigerator at −80 °C until use. Approximately 0.5 g of the leaf sample was frozen with liquid nitrogen and ground into a fine powder. Genomic DNA was extracted using the cetyltrimethyl ammonium bromide (CTAB) method [52]. The OD 260/280 ratio was measured using a Nanodrop spectrophotometer (Nano-Drop Technologies, Wilmington, NC, USA) to assess DNA purity. High-quality DNA from each sample was diluted to a concentration of 50 ng/μL with ddH2O and stored at 4 °C for use in constructing DNA bulks.

### 4.5. BSA-Seq Analysis

A total of 30 samples of each phenotype were separately selected to construct four bulked DNA samples (two papilla bulks and two non-papilla bulks) within the two F2 segregating populations. Paired-end sequencing libraries of the four parental lines and four DNA bulks were constructed. The clean sequencing data obtained were aligned to the reference genome of foxtail millet Yugu1 (T2T) using the Burrows–Wheeler Aligner (BWA) v 0.7.17 software. The alignment results were sorted into BAM (Binary Alignment/Map) format, and duplicate reads were removed using Samtools and Picard to ensure the accuracy of the subsequent analysis. SNPs and InDels were identified by the Haplotype Caller module in GATK. SNPs and InDels were annotated based on the reference genome using ANNOVAR 2018Apr16, with SNPs categorized into intergenic regions, upstream or downstream regions, and exons or introns. SNPs in coding exons were further classified as synonymous or non-synonymous SNPs. InDels located in exons were categorized based on whether they caused a frameshift. Linkage analysis between phenotypic traits and variation sites was performed using the ∆SNP-index and ∆InDel-index.

### 4.6. RNA-Seq Analyses

Transcriptome sequencing was performed on the third panicle branches or husks at four stages: the third panicle branches at the differentiation stage, the third panicle branches at the heading stage, the husks at the pollination stage, and the husks at the filling stage. Transcriptome sequencing was performed using three replicates. The cDNA library preparations and sequencing experiments were conducted by the sequencing cooperation of Novogene Co., Ltd. (Beijing, China), and the libraries were sequenced on an Illumina platform. After sequencing, we processed the raw data by filtering out low-quality reads and checking the sequencing error rate and GC content distribution to obtain high-quality clean reads. The clean reads were mapped to the Yugu1 (T2T) reference genome using HISAT2 with default parameters [53]. To quantify gene expression levels, we calculated FPKM, which normalizes for sequencing depth and gene length. Finally, we determined the FPKM-based expression values for all genes in each sample.

## 5. Conclusions

By simulating the soil environment during the sowing period of foxtail millet, it was found that the water uptake and germination of the seeds of the two accessions with papillae was significantly higher than that without papillae at the early stage of water absorption and germination. Genetic analysis indicates that the papilla density on the husks was regulated by a single dominant gene. Through BSA-Seq analysis, the candidate interval of papilla density was located on chromosome 5. Based on the BSA co-localization and expression profiling, five candidate genes were identified that may be related to the formation and development of papilla cells.

## Figures and Tables

**Figure 1 plants-14-02535-f001:**
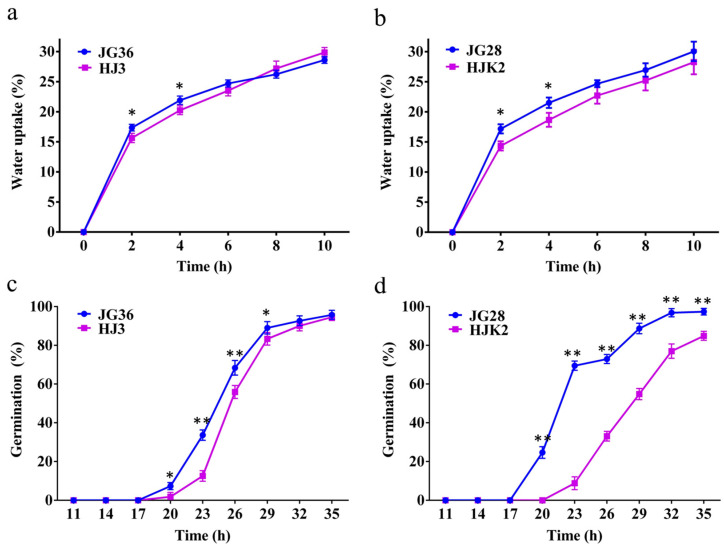
Analysis of the water uptake and germination of foxtail millet seeds during the germination: (**a**) analysis of water uptake of JG36 and HJ3 during the germination; (**b**) analysis of water uptake of JG28 and HJK2 during the germination; (**c**) analysis of germination of JG36 and HJ3; (**d**) analysis of germination of JG28 and HJK2. Note: Asterisks indicate significant differences between the varieties by Sidak’s multiple comparisons test (* *p* < 0.05, ** *p* < 0.01).

**Figure 2 plants-14-02535-f002:**
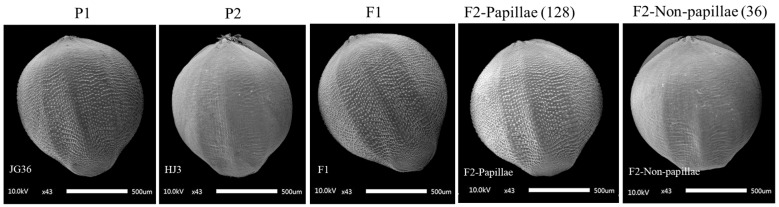
Scanning electron microscopy (SEM) images of papillae on the husks from the two parental lines (JG36 and HJ3), the F1 hybrids, and the two F2 populations.

**Figure 3 plants-14-02535-f003:**
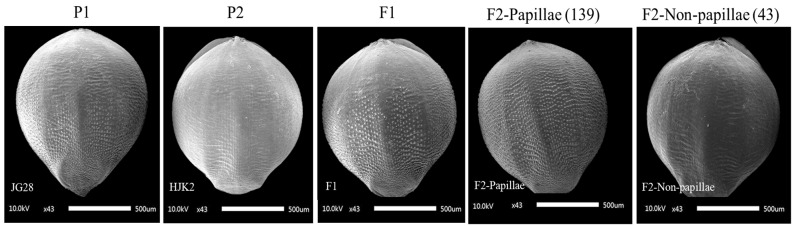
SEM images of papillae on the husks from the two parental lines (JG28 and HJK2), the F1 hybrids, and the two F2 populations.

**Figure 4 plants-14-02535-f004:**
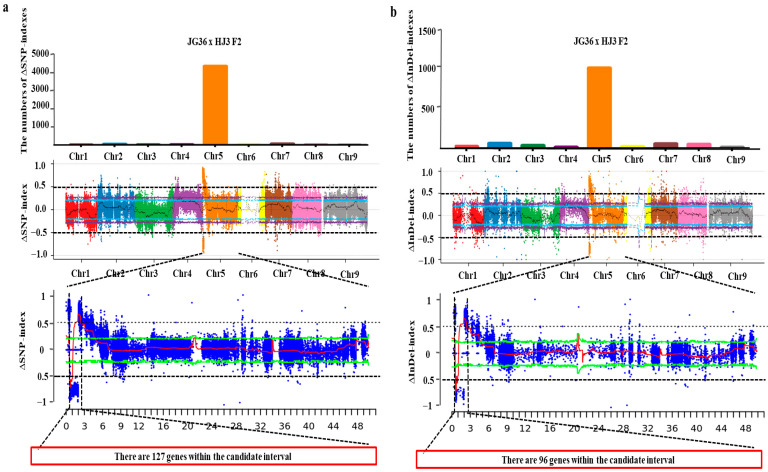
Chromosomal distribution of ΔSNP-indexes (**a**) and ΔInDel-indexes (**b**) in the JG36 × HJ3 F2 populations.

**Figure 5 plants-14-02535-f005:**
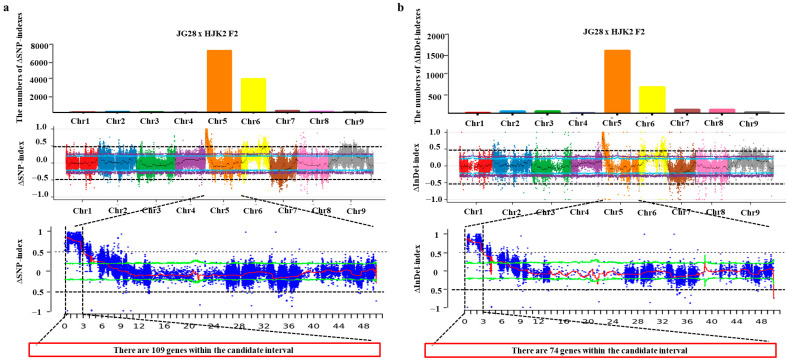
Chromosomal distribution of ΔSNP-indexes (**a**) and ΔInDel-indexes (**b**) in the JG28 × HJK2 F2 populations.

**Figure 6 plants-14-02535-f006:**
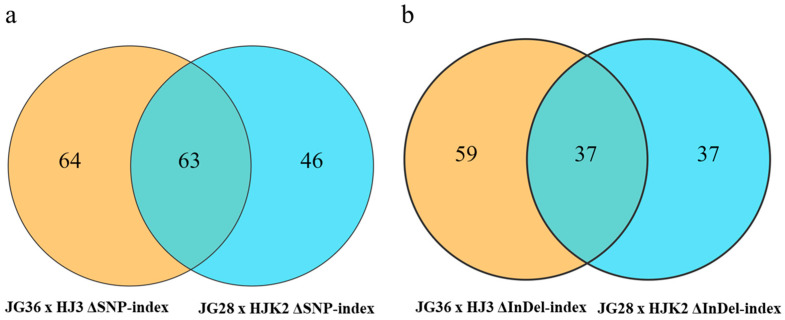
BSA co-localization analysis based on two F2 populations: (**a**) BSA co-localization analysis based on two F2 populations through ΔSNP-index association analysis; (**b**) BSA co-localization analysis based on two F2 populations through ΔInDel-index association analysis.

**Figure 7 plants-14-02535-f007:**
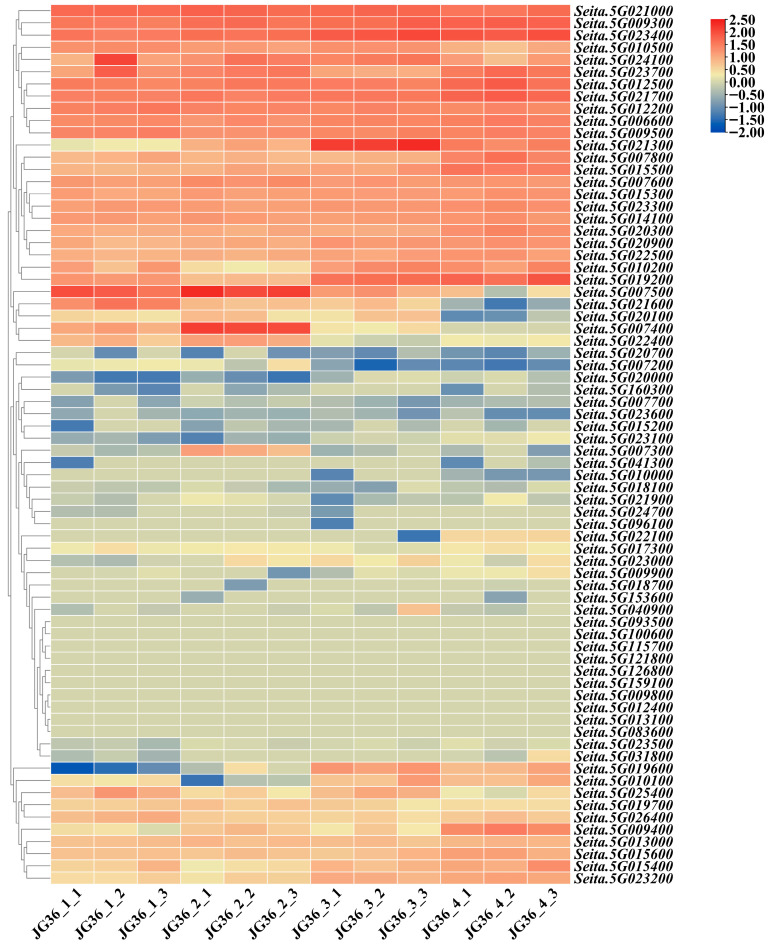
Expression patterns of candidate genes for papilla density in the husks at different developmental stages.

**Table 1 plants-14-02535-t001:** Statistical analysis of papillae on the husks in JG36 × HJ3-derived populations.

Population	Total Plants	Papilla Plants	Non-Papilla Plants	Observed Segregation Ratio	Expected Segregation Ratio	Chi-Square Test (χ2)	*p*
JG36	15	15	0	-	-	-	-
HJ3	15	0	15	-	-	-	-
F1	15	15	0	-	-	-	-
F2 (JG36 × HJ3)	164	128	36	3.6:1	3:1	0.813	0.367

**Table 2 plants-14-02535-t002:** Statistical analysis of papillae on the husks in JG28 × HJK2-derived populations.

Population	Total Plants	Papilla Plants	Non-Papilla Plants	Observed Segregation Ratio	Expected Segregation Ratio	Chi-Square Test (χ2)	*p*
JG28	15	15	0	-	-	-	-
HJK2	15	0	15	-	-	-	-
F1	15	15	0	-	-	-	-
F2 (JG28 × HJK2)	182	139	43	3.2:1	3:1	0.183	0.669

**Table 3 plants-14-02535-t003:** Quality, genome coverage, and identified variants from the re-sequencing data of the JG36 × HJ3 F2 population and the JG28 × HJK2 F2 population.

Samples	Raw Bases (Gb)	Clean Bases (Gb)	Q30 (%)	Mapped Reads	Mapping Rate (%)	Number of SNPs	Number of InDels
1-P1 (JG36, papillae)	31.7	30.7	94.50	196,594,755	96.17	2,002,017	255,095
1-P2 (HJ3, non-papillae)	31.1	29.7	94.48	192,911,451	97.29	719,803	98,418
1-papillae_bulk	53.6	51.2	94.67	331,208,868	97.00	2,113,196	283,084
1-non-papillae_bulk	53.6	52.1	94.75	337,187,736	97.04	2,092,266	281,108
2-P1 (JG28, papillae)	53.9	52.9	94.79	343,561,941	97.43	2,339,553	294,988
2-P2 (HJK2, non-papillae)	54.1	48.7	92.00	308,643,536	95.05	1,297,415	172,874
2-papillae_bulk	51.0	49.9	94.30	322,903,543	97.04	2,567,165	340,174
2-non-papillae_bulk	53.4	52.0	94.17	334,905,576	96.61	2,537,469	338,819

**Table 4 plants-14-02535-t004:** Candidate genes for foxtail millet papilla density and their homologous genes in rice.

Gene ID	Gene Names	Homologous Gene	Function Annotation of Homologous Genes
*Seita.5G007300*	*SiTIP4;2.1*	*LOC_Os01g13130*	Promoting the development of multicellular hairy bodies in tomatoes [39] and the elongation of cotton fibers [40].
*Seita.5G007400*	*SiTIP4;2.2*	*LOC_Os01g13130*	
*Seita.5G007500*	*SiTIP4;2.3*	*LOC_Os01g13130*	
*Seita.5G021600*	*SiSWEET16*	*LOC_Os03g22200*	Regulating vacuolar sugar homeostasis (if sugar transport) affects the initiation and elongation of cotton fibers (through sugar signaling) [41].
*Seita.5G022400*	*SiGH9B16*	*LOC_Os08g02220*	Participating in cell wall remodeling (such as cellulose synthesis) and promoting the thickening and strengthening of the secondary wall of cotton fibers [42].

## Data Availability

The raw sequencing data generated in this study are available in SRA (https://www.ncbi.nlm.nih.gov/sra, accessed on 12 June 2025) of NCBI with the accession numbers PRJNA1276485.

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
