# Peer review of "Identification of Candidate Genes Related to the Husk Papillae in Foxtail Millet (Setaria italica (L.) P. Beauv)"

_plants, 2025, doi:10.3390/plants14162535_

Round 1

Reviewer 1 Report

Comments and Suggestions for Authors

The overall quality of the manuscript is enough for publication with some minor correction. 

the introduction add more information about the Setaria italica 

In discussion section please add new references and more information of Setaria italica related to the study 

Author Response

Dear Reviewer: 

Plants-3744520, Identification of Candidate Genes Related to the Husk Papillae in Setaria italica Based on BSA-Seq Analysis

Thank you very much for the valuable feedback on our manuscript. We sincerely appreciate your time and effort. Please see our point by point responses to your comments and the revisions we have made to improve the manuscript.

  1. the introduction add more information about the Setaria italica

Reply: We have now integrated more references and information into the Introduction (Line 46-51) and Discussion sections (Line 259-270, Line 304-307).

  1. In discussion section please add new references and more information of Setaria italica related to the study

Reply: In the revised manuscript, we have now added more references and information of Setaria italica related to the study in the discussion section (line 259-270, Line 304-307).

Many thanks again for your comments and suggestions. 

Reviewer 2 Report

Comments and Suggestions for Authors

The minor comments must address to improve the quality of the proposed work and manuscript.

  1. The abstract is designed well.
  2. Introduction, lines 45-47, add more references for the supporting of the germination, only single reference is not good enough to support the given statement of Water uptake.
  3. Lines 36-85, the induction is written well but the authors cited limited reference. I strongly recommend reviewing the recently published article and improving the introduction section and cite them as appropriately.
  4. Lines 88-91, Write the full form of the abbreviation used at first time and continue to use the abbreviation., e.g., define JG36, JG28, HJ3, HJK2 etc.
  5. The results section is presented well.
  6. Lines 93-96, please clarify how the water uptake has been determined. Is it based on visual observation or used some technique to be determined it.
  7. It is strongly suggested to change section 4 to section 2 and accordingly update the section. Otherwise, it may be difficult to understand the results provided in this article. I understand its journal style, but the reader or new researcher can’t catch up with the proposed work.
  8. Line 285 – 291, add the manufactures or seed bank details (such as name, location) for the collected foxtail millet varieties.
  9. Line 293, what method is used for the water content determination for the dry seed (e.g. KF, LOD or others)?
  10. Lines 316, explain more about the CTAB method?
  11. Line 318, provide more details about the Nanodrop spectrophotometer and method.
  12. The section must improve by adding the key test results.

Author Response

Dear Reviewer:

Plants-3744520, Identification of Candidate Genes Related to the Husk Papillae in Setaria italica Based on BSA-Seq Analysis

Thank you very much for the valuable feedback on our manuscript. We sincerely appreciate your time and effort. Please see our point by point responses to your comments and the revisions we have made to improve the manuscript.

  1. Introduction, lines 45-47, add more references for the supporting of the germination, only single reference is not good enough to support the given statement of Water uptake.

Reply: We have added more references in the “Introduction” section (line 46).

  1. Lines 36-85, the induction is written well but the authors cited limited reference. I strongly recommend reviewing the recently published article and improving the introduction section and cite them as appropriately.

Reply: We have reviewed the recently published papers and cited them as appropriately in the “Introduction” section (line 46-51).

  1. Lines 88-91, Write the full form of the abbreviation used at first time and continue to use the abbreviation., e.g., define JG36, JG28, HJ3, HJK2 etc.

Reply: We appreciate the reviewer’s attention to the details. In the “Results” section (line 93-94), we have added the full forms of the abbreviations JG36, JG28, HJ3 and HJK2.

  1. Lines 93-96, please clarify how the water uptake has been determined. Is it based on visual observation or used some technique to be determined it.

Reply: We have added the method for determining the water uptake in the “Materials and Methods” section (line 319-326).

  1. Line 285 – 291, add the manufactures or seed bank details (such as name, location) for the collected foxtail millet varieties.

Reply: We have added the related informations as suggested, in the “Materials and Methods” section (line 312-313) and “Results” section (line 93-94).

  1. Line 293, what method is used for the water content determination for the dry seed (e.g. KF, LOD or others)?

Reply: We have added the method that was used for the water content determination for the dry seed in the “Materials and Methods” section (line 319-320).

  1. Lines 316, explain more about the CTAB method?

Reply: The CTAB method used for extracting genomic DNA in the experiment was based on the reference cited below.

Doyle, J.; Doyle, J.L. Isolation of Plant DNA from fresh tissue. Focus 1990, 12, 13–15.

  1. Line 318, provide more details about the Nanodrop spectrophotometer and method.

Reply: The instrument used for detecting the purity of DNA is the Nanodrop spectrophotometer (Nano-Drop Technologies, Wilmington, NC, USA).

  1. The section must improve by adding the key test results.

Reply: The key test results of the BSA-Seq analysis are presented in section 2.3 (Figures 4 and 5).

Many thanks again for your comments and suggestions.

Reviewer 3 Report

Comments and Suggestions for Authors

Abstract

Setaria italica is a cereal with a high protein content in its seeds (used in human and animal feed) and greater drought tolerance compared to other grain crops, making it particularly suitable for cultivation in arid and semi-arid regions, such as northern China. In dryland cultivation in arid regions, rapid germination is critical for the successful establishment of the plant population. In the germplasm of Setaria italica, accessions with varying sizes, densities, and arrangements of papillae on the epidermis of their husks have been described. Literature suggests that these papillae are associated with water uptake by the seed, but this relationship has not yet been studied in Setaria italica, nor is the genetic basis of the trait known in the species. The objective of this study was to investigate the relationship between the papillae present on the epidermis of the seed husks of Setaria italica and water uptake by the seeds, as well as to clarify the genetic basis of this trait. The study found that the water uptake rate observed during germination was higher in seeds with papillae compared to those without, demonstrating that the presence of papillae promotes greater water absorption by the seeds. Phenotypic analysis of F1 and F2 populations obtained from the cross of accessions with and without papillae on the seed husks showed that the trait is monogenic and the gene conferring papillae presence being dominant. A BSA-Seq analysis using four parental lines and bulks of individuals with and without papillae from the segregating F2 population was conducted, and a region significantly associated with the genetic control of the trait was identified on chromosome 5, where 72 genes were located. From gene expression analysis and functional annotation, five genes putatively related to papilla density were identified, encoding aquaporins, a fructose transporter, and a glycoside hydrolase. It is concluded that these genes, individually or in combination, are involved in regulating papilla density on the seed husks of Setaria italica. These findings represent an advance in understanding the relationship between the presence of papillae on the seed husks of Setaria italica and water uptake by the seeds, and reveal the genetic control of this trait. This knowledge will contribute to defining breeding strategies to develop cultivars with higher papilla density on the seed husks, and consequently, better germination, which is particularly important for dryland cultivation in arid regions. Germination efficiency is crucial for achieving a population density that contributes to high and stable productivity. The candidate genes putatively associated with the regulation of papilla density on the seed husks, and consequently water uptake by the seeds, provide insights for further research to clarify the water uptake process during seed germination.

Positive Points:

The study is well-justified, and the results provide significant scientific and practical contributions to the species.

The methodological procedures and software used in the genetic-statistical analyses are robust and suitable for the study's objectives and to test the hypothesis.

The conclusions are consistent with and supported by the obtained results, and are properly discussed."

The results clarify the relationship between the presence of papillae on the seed husks and water uptake by seeds.

The manuscript presents previously unknown information on the genetic basis of papillae presence on the seed husks.

Candidate genes putatily associated with papillae presence on the seed husks are identifies.

Negative Points and suggestions:

The keywords are not appropriate, they repeat terms already present in the title.

Author Response

Dear Reviewer:

Plants-3744520, Identification of Candidate Genes Related to the Husk Papillae in Setaria italica Based on BSA-Seq Analysis

Thank you very much for the valuable feedback on our manuscript. We sincerely appreciate your time and effort. Please see our point by point responses to your comments and the revisions we have made to improve the manuscript.

  1. The keywords are not appropriate, they repeat terms already present in the title.

Reply: We have appropriately modified the title of the paper based on the research content. 

Many thanks again for your comments and suggestions.

Reviewer 4 Report

Comments and Suggestions for Authors

The manuscript deals with combined BSAseq and transcriptomic RNAseq analysis of  husk papilae on foxtail millet seeds. It is sound study, I have only minor points to improve the manuscripts.

I suggest to modify the title, as the study is based on combination of BSAseq and RNAseq as mentioned. Actually it might perhaps be just like this: "Identification of Candidate Genes Related to the Husk Papillae  in foxtail millet (Setaria italica (L.) P. Beauv)".

Introduction, line 75 reference 30, it would be good to include the species name before the genome, it is on Rice.

Figure 1. I must say based on figure it does not look to convincing to see the statistical difference in water uptake, but I believe the data, what is however more important that later on this difference between genotypes is not present and even might be in oppositive (Fig. 1a) any comments to this? the behaviour 2 to 4h is early, but when is the actual seed germination?

Section 2.2 (lines 109-126) how it is with crosses in term of the parents, which one was used at mother ? this is important in term of F1 and the trait, since the seed coat/husk is of maternal origin. Was perhaps done some recirpocal crosses? Table 1 and 2, would be good to include observed and expected values, not just the chi square only.

BSAseq is impressive in term on so narrow interval identification. Seems to be lucky.

Section 2.5 on expression. Would be usefull to include more detailed genes description into the supplementary material.

Section 2.6 "Functional Prediction of Candidate Genes for Papilla Density" it is not actually about function, as this part would require further testing. It is simply continuation of the previous section to which it could be linked. What is unclear how those 5 candidate genes were selected ? it should include the expression level and also indicate the respective chromosomal positions.

Discussion part would be nice to include some information about the occurence/frequence of the husk papillae within the germplasm, which could also suggest possible selection. How this trait is present in wild progenitor?

Discussion would benefit from expanision providing more comparison to seed surface structural arrangement and the germination or the camouflage or seed protection from predation as well as it might aid seed dispersion via zoochory. Please include this, there are some available references to this including foxtail millet.

Abstract lines 22, improve the wording of the sentence, specifically time of the seed water uptake.

Comments on the Quality of English Language

There are some sentences which would benefited from grammar and style improvement, should be detected by classical text editing software like Grammarly for example.

Author Response

Dear Reviewer

Plants-3744520, Identification of Candidate Genes Related to the Husk Papillae in Setaria italica Based on BSA-Seq Analysis

Thank you very much and the reviewer for the valuable feedback on our manuscript. We sincerely appreciate your time and effort. Please see our point by point responses to your comments and the revisions we have made to improve the manuscript.

  1. I suggest to modify the title, as the study is based on combination of BSAseq and RNAseq as mentioned. Actually it might perhaps be just like this: "Identification of Candidate Genes Related to the Husk Papillae in foxtail millet (Setaria italica (L.) P. Beauv)".

Reply: We have modified the title based on your suggestion.

  1. Introduction, line 75 reference 30, it would be good to include the species name before the genome, it is on Rice.

Reply: We have included the species name in the “Introduction” section (line 78).

  1. Figure 1. I must say based on figure it does not look to convincing to see the statistical difference in water uptake, but I believe the data, what is however more important that later on this difference between genotypes is not present and even might be in oppositive (Fig. 1a) any comments to this? the behaviour 2 to 4h is early, but when is the actual seed germination?

Reply: Foxtail millet exhibits exceptional water economy during germination, requiring only 26% of seed weight in water uptake for successful germination. As showed in the following literature:

Gu, S. Relationship between foxtail millet growth and environmental factors. In Foxtail millet cultivation and production in China, Gu, S., Chinese Agricultural Press: Beijing, China, 1987, 63-71.

In this study, the water uptake by foxtail millet seeds was extremely rapid, with water content (the original water content of the seeds before sowing plus the water uptake after sowing) reaching approximately 30% (Table S1 & 5) within just 2 hours after sowing. Seeds of accessions with papillae exhibited significantly higher water uptake than those without papillae at both two and four hours after sowing. This accelerated hydration enables timely acquisition of sufficient water for germination, a critical advantage in arid regions like northern China, where soil moisture rapidly diminishes after rainfall due to intense evaporation driven by strong sunshine and winds during the sowing season.

In the Results section, we have included the results of germination rates(line 102-105). After 20 hours of sowing, JG36, HJ3, and JG28 had already germinated. The germination rate of JG36 and JG28 was significantly higher than that of HJ3 and HJK2, respectively, twenty to twenty-nine hours after sowing (Figure 1 and Table S1).

  1. Section 2.2 (lines 109-126) how it is with crosses in term of the parents, which one was used at mother ? this is important in term of F1 and the trait, since the seed coat/husk is of maternal origin. Was perhaps done some recirpocal crosses? Table 1 and 2, would be good to include observed and expected values, not just the chi square only.

Reply: The HJ3 and HJK2 were used as the maternal parent for the two sets of crossing. We agree that it would be better if recirpocal crosses were carried out. We incorporated the observed and expected values as suggested into these tables as shown in Tables 1 and 2. 

  1. Section 2.5 on expression. Would be usefull to include more detailed genes description into the supplementary material.

Reply: We have included the expression levels of all genes at different developmental stages of the husk in the Table S3 (revised Supplementary Materials).

  1. Section 2.6 "Functional Prediction of Candidate Genes for Papilla Density" it is not actually about function, as this part would require further testing. It is simply continuation of the previous section to which it could be linked. What is unclear how those 5 candidate genes were selected ? It should include the expression level and also indicate the respective chromosomal positions.

Reply: We fully agree with your comments and will conduct further testings to verify gene function in the subsequent research. Additionally, the five candidate genes were selected through the analysis of gene expression differences. During the third stage of husks development, that is, the pollination stage, the papilla cells on the husks have been fully differentiated and formed; the candidate genes regulating papilla formation should be highly expressed during the first two stages. In the analysis of the expression differences of 72 genes, we discovered that five candidate genes were found to be highly expressed in the husks of the first two stages. Furthermore, we have included the expression level and the chromosomal positions of the candidate genes in the Table S4 (revised Supplementary Materials).

  1. Discussion part would be nice to include some information about the occurence/frequence of the husk papillae within the germplasm, which could also suggest possible selection. How this trait is present in wild progenitor?

Reply: Thank you very much for your suggestions. The investigation of the occurence/frequence of the husk papillae among the germplasms of S. italica and its wild relative S. viridis could reveal if the trait is selected during domestication. We have added related discussion (line 304 - 307).

  1. Discussion would benefit from expanision providing more comparison to seed surface structural arrangement and the germination or the camouflage or seed protection from predation as well as it might aid seed dispersion via zoochory. Please include this, there are some available references to this including foxtail millet.

Reply: We fully agree with your view that the seed surface structural arrangement of foxtail millet not only relates to water uptake and germination, but may also play a role in preventing bird feeding and seed dispersal. We have added related discussion (line 259-270). 

  1. Abstract lines 22, improve the wording of the sentence, specifically time of the seed water uptake.

Reply: Thank you very much for your suggestions. We have modified the abstract.

Many thanks again for your comments and suggestions.
